# Immediate Effects of Anti-Spastic Epidural Cervical Spinal Cord Stimulation on Functional Connectivity of the Central Motor System in Patients with Stroke- and Traumatic Brain Injury-Induced Spasticity: A Pilot Resting-State Functional Magnetic Resonance Imaging Study

**DOI:** 10.3390/biomedicines11082266

**Published:** 2023-08-14

**Authors:** Larisa Mayorova, Margarita Radutnaya, Maria Varyukhina, Alexey Vorobyev, Vasiliy Zhdanov, Marina Petrova, Andrey Grechko

**Affiliations:** 1Laboratory of Physiology of Sensory Systems, Institute of Higher Nervous Activity and Neurophysiology of the Russian Academy of Sciences, 117485 Moscow, Russia; 2Federal Research and Clinical Center of Intensive Care Medicine and Rehabilitology, 107031 Moscow, Russia; 3Department of Anesthesiology and Resuscitation with Medical Rehabilitation Courses, Peoples’ Friendship University of Russia (RUDN University), 117198 Moscow, Russia

**Keywords:** spasticity, upper motor neuron, resting-state fMRI, functional connectivity, spinal cord stimulation, antispastic therapy

## Abstract

Objective: Spinal cord stimulation (SCS) is one approach to the potential improvement of patients with post-stroke or post-traumatic spasticity. However, little is known about whether and how such interventions alter supraspinal neural systems involved in the pathogenesis of spasticity. This pilot study investigated whether epidural spinal cord stimulation at the level of the C3–C5 cervical segments, aimed at reducing spasticity, alters the patterns of functional connectivity of the brain. Methods: Eight patients with spasticity in the right limbs as a result of left cerebral hemisphere damage (due to hemorrhagic and ischemic stroke or traumatic and anoxic brain injury) were assessed with fMRI immediately before and immediately after short-term (1 to 6 days) test cervical epidural SCS therapy. Eight demographically and clinically comparable patients with spasticity in the right extremities due to a left hemisphere ischemic stroke and brain injury who received conventional therapy were examined as a control group. All patients also had paresis of one or two limbs and hyperreflexia. Results: After the SCS therapy, there were three main findings: (1) higher functional connectivity of the brainstem to the right premotor cortex and changes in functional connectivity between cortical motor areas, (2) increased functional connectivity between the right and left lateral nodes of the sensorimotor network, and (3) a positive correlation between decreased spasticity in the right leg and increased functional connectivity within the right hemisphere sensorimotor cortex. All these changes in functional connectivity occurred with a statistically significant decrease in spasticity, as assessed using the modified Ashworth scale. The control group showed no decrease in spasticity or increase in functional connectivity in any of the seeds of interest. On the contrary, a decrease in functional connectivity of the brainstem and right postcentral gyrus was observed in this group during the observation period. Conclusions: We were thus able to detect intrinsic brain connectivity rearrangements that occurred during spasticity mitigation following short epidural SCS therapy. Significance: The clinical results obtained confirmed the efficacy of short-term anti-spastic SCS therapy. The obtained data on functional rearrangements of the central motor system may shed light on the mechanism of antispastic action of this procedure.

## 1. Introduction

Post-stroke spasticity occurs in 37.5–45% [1,2] and 19–57.4% [1,3,4] of cases in the acute and subacute stages after a stroke, respectively. At 6 months after a stroke, spasticity develops in 42.6–49.5% of cases [2,5], and at one year, in 35–57.4% [1,2,3]. Spasticity after a traumatic brain injury is even more common, occurring in 59–70% of cases one year after the trauma [6] and in up to 89% of cases in the longer term [7].

Spasticity, i.e., increased muscle tension, is considered a classic consequence of an upper motor neuron (UMN) lesion [8] following damage to the central nervous system, including trauma and stroke. Clinically, spasticity manifests as muscle resistance to passive stretching/extension and is often accompanied by manifestations such as clasp-knife rigidity, clonus, increased tendon reflexes, and muscle spasms [9]. Pathophysiologically, spasticity is the form of hypertonia due to velocity- and length-dependent increases in the exaggeration of the muscle stretch reflex. The increase in the reflex is a consequence of abnormal activity of muscle spindles and abnormal spinal processing of proprioceptive inputs [8,10,11,12,13,14]. Both supraspinal and spinal motor control structures play a role in the development of spasticity/exaggeration of the muscle stretch reflex and spastic paresis [9,15,16].

Supraspinal components of motor control include the cerebral cortex (frontal motor area forming corticospinal (pyramidal) pathways, premotor and supplementary motor cortices, prefrontal cortex projecting to premotor and supplementary motor areas, parietal cortical areas, and association areas (visual, tactile, auditory)), basal ganglia (striatum, pallidum, substantial nigra, subthalamic nucleus), cerebellum, and brainstem. Downward influences from supraspinal structures (of the motor system) control the spinal reflexes, converging with peripheral afferents into an interneuron pool projecting to motoneurons. An imbalance of descending inhibitory and facilitatory influences on muscle stretch reflexes is thought to be the cause of spasticity [17]. In humans, three major tracts are known to transmit these downward influences: corticospinal, reticulospinal (RST) [18,19], and vestibulospinal tracts (VST) [18]. Even though pyramidal cells influence muscle tone, an isolated corticospinal tract injury does not produce spasticity [11,20,21], but isolated damage to pyramidal cells and their axons is extremely rare and is more often combined with damage to fibers responsible for spasticity from premotor and supplementary motor areas [9]. The lateral RST originating in the reticular formation in the medulla receives extensive convergent inputs from the primary and supplementary motor cortex bilaterally [21,22] and directs signals to the spinal motoneuron. The medial VST receives signals from subcortical structures and the lateral vestibular nucleus and directs excitatory effects to spinal stretch reflexes [23].

Regarding spinal plasticity mechanisms, these are complex plastic rearrangements secondary to an imbalance of excitatory and inhibitory descending influences on the intraspinal neural network, which are described in detail in [9].

The treatment of spasticity has traditionally included medication, surgery, and neurosurgery. When spasticity occurs, combined medication is the first choice. Almost all drugs used to treat spasticity have undesirable side effects, ranging from adverse effects on neural recovery [22] and cognitive decline [23] to atrophy of the muscles involved [24] and impaired diaphragm contractile function [25]. Selective dorsal rhizotomy has been successfully used in the treatment of spasticity and the improvement of lower limb function in spastic diplegia. In general, however, the efficacy and success of this treatment remain unpredictable and are individual for each patient [26]. Deep brain stimulation is used in the treatment of dystonia in combination with spasticity and results in a significant improvement in the quality of life and disability that persists for 3–5 years after the procedure [27]. There are complications of deep brain stimulation associated with inappropriate stimulator placement and infection. In contrast with oral administration, intrathecal administration of baclofen (ITB) causes fewer adverse reactions and requires a lower dosage, but toxic doses of baclofen cause loss of consciousness and respiratory depression, while rapid baclofen withdrawal can lead to seizures, hallucinations, and hyperthermia with possible multi-organ failure due to rhabdomyolysis [28]. According to a meta-analysis by Korzhova J. et al. (2018), a clinically and statistically significant effect of transcranial magnetic stimulation (TMS) has only been shown in a group of patients with acquired spasticity from brain stem and spinal cord lesions. There was no statistically significant effect of using TMS to treat spasticity after a stroke [29].

Finally, evidence, although incomplete, suggests that spinal cord stimulation (SCS) with implantable electrodes may be effective in reducing spasticity. Epidural spinal cord stimulation has been used to treat neuropathic pain for about 50 years [30]. More recently, it has become clear that SCS is not limited to its anti-pain effects, but also mediates improvements in motor, sensory, and bladder function [31,32,33,34,35,36]. The use of spinal cord stimulation to reduce spasticity has been discussed for quite a long time [37]. Positive clinical results of SCS application in the form of spasticity reduction in patients with multiple sclerosis [38,39], movement disorders [32,40,41], stroke after-effects [42], cerebral palsy, and spasticity caused by spinal injury [43,44,45,46], treatment of hypoxia of the brain [47] are documented. In many cases, the effect lasted long enough. More details can be found in the literature reviews of Gybels and Van Roost (1991) and Nagel et al. (2017) [37,48]. SCS therapy for the treatment of spasticity was superseded by ITB, but there are several undeniable advantages of SCS that cannot be overlooked. These advantages are mainly due to the lack of a pharmaceutical agent: first, drugs always have side effects and cases of intolerance, and second, drug titration is much more difficult than device calibration [37]. It should be noted that like other invasive procedures, epidural SCS can have several complications, such as biological- and hardware-related problems. The most common problem reported in the first type is infection; in the second type, lead migration is a common issue [49]. Interest in epidural spinal cord stimulation in spasticity treatment is gradually returning.

The mechanisms of the therapeutic effect of spinal cord stimulation for pain relief are currently unclear. The authors hypothesize that SCS works by blocking pain signal transmission, interfering with the pain pathway, activating the opioid pathway, stimulating the locus coeruleus system, and regulating GABA energy [50]. Little is known about the supraspinal mechanisms of action of spinal cord stimulation in the treatment of pain syndrome, and almost nothing in the treatment of spasticity. There is some information on the effects of SCS on sensorimotor and insular cortical activity in the treatment of neuropathic leg pain (with appropriate distal electrode placement) [51] and on enhancing the synchronization of sensory and limbic areas [52]. These studies differed in experimental design, but both had scans with the stimulator switched off and on (located at the level of the thoracic vertebrae). In neither fMRI study was the effect of preconditioning from SCS excluded (SCS was performed some time before the scan), and thus, it cannot be said that the effect of SCS is limited to the observed changes. In addition, with most of the devices available on the market, mounting the system at the cervical spinal cord level would not technically allow for such a study to be realized. These results and the results of other neuroimaging studies [53], including a systematic review involving, amongst others, 10 studies using hemodynamic imaging techniques [54], suggest that SCS is able to modulate supraspinal neuronal activities, which downregulate the negative affective components of pain. It should be noted, however, that the evidence level of these studies is low, theories about the therapeutic mechanism cannot be formed, and much work remains to be done [54,55]. For stroke- or TBI-induced spasticity that is due to an upper motor neuron lesion, clarifying the presence of supraspinal or cortical mechanisms of SCS is particularly relevant to understanding the existence of therapeutic mechanisms of action of the procedure. However, we were only able to find one paper that discussed this issue. A case report was made for a patient with anoxic brain injury, and right-sided spasticity in the arm and leg decreased after SCS. Single photon emission computed tomography (SPECT) showed increased cerebral blood flow in the basal ganglia, thalamus, brain stem, and cerebellum during stimulation [47].

In general, elucidating the supraspinal mechanisms of the therapeutic effect of stimulation, and therefore determining the presence of a positive effect of SCS therapy in spasticity syndrome, is complicated by several factors: (1) When the stimulator is placed at the cervical spinal cord level, fMRI cannot be performed with most commercially available devices; the situation is further complicated by the fact that the system is usually fitted on a permanent basis, which does not allow for observation of the effect of SCS on functional connectivity using repetitive resting-state fMRI, which is the only technique currently available that provides spatially high-resolution data that includes deep structures. (2) Monitoring the placebo effect is difficult from an ethical point of view. (3) There is insufficient data on the pathophysiology of spasticity, on the fingerprints of spasticity on the pattern of brain activity, and on adaptive neuroplastic rearrangements associated with spasticity mitigation. Problem (1) can be partially solved if the electrodes are implanted for a certain period, e.g., as in test epidural spinal cord stimulation. Therefore, we examined changes in intrinsic brain connectivity after anti-spastic epidural cervical spinal cord stimulation in patients with spasticity syndrome after left-hemisphere brain damage. We hypothesized that following short (about a week) epidural spinal cord stimulation in patients after stroke or trauma, intrinsic brain connectivity rearrangements amid elimination of spasticity could be (1) present and detectable with RS-fMRI, and (2) these changes could reflect the therapeutic effect of the procedure. As this study is a pilot, we did not aim to test hypotheses about the safety, efficacy, and effectiveness of SCS.

This controlled pilot study described rearrangements of brain functional connectivity in acquired spasticity after a short course of anti-spastic epidural cervical SCS therapy and the association of these changes with clinically observed reductions in spasticity.

## 2. Materials and Methods

### 2.1. Participants

Patients admitted to our hospital between January 2020 and March 2023 were included. Patients were recruited by a group of surgeons and neurologists into two groups: the main group (SCS therapy) and the control group (standard therapy). The inclusion criteria were (1) first-ever stroke, (2) aged between 19 and 80 years, and (3) left-sided brain injury associated with right-sided spasticity in one or both limbs. The exclusion criteria were as follows: (1) any premorbid history of psychiatric or neurological pathology and (2) the presence of any contraindication for MRI. The main group included eight patients (3 women, 5 men; mean age (SD) 41 (15.78) years) with right-sided paresis and spasticity. In 1 patient, the original cause of spasticity was anoxic brain injury; in 4 patients, spasticity developed due to stroke; and in 3 patients, spasticity developed due to TBI. In 4 patients, the duration of illness ranged from 190 to 3794 days, 2 patients were in the subacute stage after a stroke (22, 81 days), and the remaining 2 patients were in the chronic phase of TBI (65 days). All patients had spasticity in both right limbs, except one who had it only in the arm. The control group consisted of 8 persons (2 women, 6 men; mean age (SD) 56.5 (11) years) with right-sided paresis and spasticity due to ischemic stroke, except one patient with TBI (Table 1). All patients also had paresis of one or two limbs and hyperreflexia. 

Patients in the main and control groups did not differ significantly in terms of their disease duration, lesion volume, etc. A comparative analysis of the groups is shown in Table 2.

The superposition of lesion localization in the patient groups is shown in Figure 1. 

### 2.2. Ethics Statement

The study was conducted in accordance with the principles of biomedical ethics formulated in the 1964 Helsinki Declaration and its subsequent updates and was approved by the local bioethical committee of the Federal Research and Clinical Center of Intensive Care Medicine and Rehabilitology (Moscow) (protocol no. 1/19/12). All participants submitted voluntary written informed consent signed by the participant themselves for the publication of any potentially identifiable images or data included in this article. 

### 2.3. Study Design

Patients of the main group received a test epidural antispastic stimulation system at the C3–C5 level for a period of 1 to 6 days. The group underwent two resting-state fMRI scans: (1) baseline assessment, which was performed the day before or on the day of the start of SCS therapy, and (2) post-treatment, which was performed on the last day of the SCS therapy or the day after. Patients in the control group had standard therapy and underwent two resting-state fMRI scans with the same time interval as in the main group. The flowchart of patient selection is presented in Figure 2.

### 2.4. Spasticity Assessment and Analysis

Initially, spasticity syndrome was diagnosed by the attending physician. As part of the study, spasticity was assessed using the modified Ashworth scale (MAS) by two independent specialists at the hospital before and after the temporary/test spinal cord stimulator was placed. The neurologists were not told whether the patient was from the main or control group. The MAS provides a quantitative assessment of resistance to passive movements. The MAS is simple, instrument-free, easy, and quick to perform and has been used in many studies [56,57]. Spasticity was assessed twice: on the same days as the pre-operative fMRI and the post-operative fMRI. In the control group, spasticity was assessed on the same days as the fMRI. Data analysis was carried out using SPSS 23.0 (IBM Corp., Armonk, NY, USA). Wilcoxon signed-rank tests were run to examine differences across time (baseline versus post-therapy) for upper and lower spastic extremities. Given the small sample size, corrections for multiple comparisons were not used; results were considered significant at *p* < 0.05. Spasticity assessment data for the paretic limbs of both groups of patients at baseline are presented in Table 1. 

### 2.5. SCS Therapy: Temporary Electrode Installation and Stimulation Protocol

According to the clinical manifestations, the aim of achieving a certain cervical level was the objective that guided the installation of the electrodes. A trial epidural electrode was implanted in the C3 to C5 area under general anesthesia. The patient was positioned on the left side and the head was secured using a Mayfield cranial stabilization system. The procedure included parasagittal transcutaneous puncture of the epidural space in the Th1–Th2 area with a Tuohy needle and insertion of the electrode in C3–C5 under intraoperative X-ray monitoring. An eight-pin monoaxial electrode (Octrode^TM^, Abbott Medical, Plano, TX, USA) was used. A test pulse generator (Abbott Medical, Plano, TX, USA previously St. Jude Medical, Inc., St. Paul, MN, USA) was then connected to the distal part of the electrode to check the impedance. The peripheral part of the electrode was fixed on the skin. The entire procedure took fifteen minutes and was performed by a qualified neurosurgeon. A CT scan of the neck was performed after surgery to confirm the position of the electrode and rule out surgical complications (an example of an electrode position is shown in Figure 3). There were no cases of incorrect placement of the stimulator. 

The stimulator was connected on the 1st day after surgery. Stimulation options were isolated tonic mode, “Burst”, or combinations thereof. The decision to choose a mode was made based on the presence of the clinical effect. With tonic mode, the pulse width was set to 200 µs, and with “Burst” mode, the pulse width was set to 1000 µs (5 pulses per burst). The stimulation frequency was 40 Hz for both modes. For testing, the initial current was set at 1 mA, and then it was gradually increased to the value at which the limb tremor appeared. Then, the current was decreased by 1 mA. This was the value of the current at which the stimulation was performed.

All patients underwent physical therapy and passive exercises during stimulation. None of the patients experienced any complications during pre- or postoperative intervention or while undergoing stimulation.

### 2.6. Functional MRI Scanning Parameters

Resting-state functional and anatomical images were acquired a day before the SCS system installation and right after removal using a 1.5 T Siemens Essenza (Siemens, Ltd., Munich, Germany) with an eight-channel head coil. Each resting-state functional run consisted of 300 T2*-weighted echoplanar images (EPIs). The imaging parameters were as follows: a 3.9 × 3.9 mm in-plane voxel size, covering the whole brain volume with 4.0 mm slices, an interslice gap of 0.8 mm, a repetition time (TR) of 3670 ms, an echo time (TE) of 70 ms, and a 64 × 64 matrix. In addition to the functional images, we collected a high-resolution T1-weighted anatomical scan for each participant (192 slices, resolution 1 × 1 × 1 mm, TR = 10 s, TE = 4.76 ms, 256 × 256 acquisition matrix). Patients were instructed to relax, lie still with their eyes closed, and not to think about anything in particular. Patients were not sedated during the scanning procedures.

### 2.7. Resting State Data Preprocessing Pipeline 

The data were processed using the CONN functional connectivity toolbox (http://www.nitrc.org/projects/conn, accessed on 23 February 2023), version 19c, and SPM12 (http://www.fil.ion.ucl.ac.uk/spm, accessed on 23 February 2023). The preprocessing procedure consisted of the realignment of functional images (motion correction), slice timing correction, co-registration, segmentation of structural data, normalization into standard stereotactic Montreal Neurological Institute (MNI) space, outlier detection/scrubbing using the artifact detection tool (ART) (http://www.nitrc.org/projects/artifact_detect, accessed on 23 February 2023), and spatial smoothing with a Gaussian kernel of an 8 mm full width at half maximum. Denoising was performed by removing the following confounders using linear regression: the blood-oxygen-level-dependent (BOLD) signal from the white matter and CSF masks (5 principal components of each signal), scrubbing (the number of regressors corresponded to the number of identified invalid scans), and motion regression (12 regressors: 6 motion parameters and 6 first-order temporal derivatives). In addition, manually created lesion masks in the MRIcron toolbox [58] were included in the denoising step to regress out any lesion-related signal. The resulting signals were band-pass-filtered at 0.008–0.12 Hz. 

### 2.8. Functional Connectivity Analysis

To test the hypothesis of increased cortical influences on reticular formation structures inhibiting α-motoneurons, seed-to-voxel analysis was used. Connectivity maps were created by calculating the correlation between the average time series in the brainstem region and all other voxels in the brain. The correlation maps were subsequently normalized to z-coefficients using the Fisher-Z transformation. The other seeds of interest, given the pathogenesis of spasticity, were the cortical areas associated with motor function: the precentral and postcentral gyri, which are supplementary motor areas that, like brainstem areas, are contained in the CONN package’s atlas. We performed a seed-based functional connectivity analysis using a mixed-design analysis of variance (RM-ANOVA 2 × 2) with one between-subjects factor (groups) and one within-subjects factor (therapy). All results are presented using a statistical significance threshold at the cluster level of *p* < 0.05 and adjusting for multiple comparisons (FDR). Paired comparisons were used to compare the functional connectivity before and after the SCS (contrast post-SCS minus pre-SCS) in the main group (as a nonsense covariate, models included the number of days of stimulation). A similar post hoc analysis was performed on the control group to compare the functional connectivity in the first and second control points. Regions with significant differences (*p* < 0.01) were identified and adjusted for multiple comparisons (FDR) at the cluster level *p* < 0.05.

For ROI-to-ROI analysis, in addition to the above 7 areas from the implemented atlas, we included in the list of ROIs the sensorimotor network consisting of three nodes: the right and left lateral components and the upper/central component. The results were considered significant at *p* < 0.05 at the cluster level adjusted for multiple comparisons (FDR).

We also investigated the presence of a correlation between the change in hand and leg spasticity (e.g., post-SCS spasticity minus pre-SCS spasticity according to the MAS) and the change in functional connectivity (e.g., post-SCS whole-brain functional connectivity minus pre-SCS whole-brain functional connectivity). Results were considered significant at *p* < 0.05 with corrections for multiple comparisons.

## 3. Results

### 3.1. Spasticity Elimination

A Wilcoxon signed-rank test was conducted to compare the spasticity in the hand and leg, as assessed using the modified Ashworth scale at baseline and after the SCS therapy course. In the main group, there was a significant difference in the scores for hand spasticity at baseline (M = 2.88, SD = 0.83) and after the SCS therapy (M = 1.75, SD = 1.04); Z(8) = 2.44, W = 28.00, *p* = 0.0074. For the leg, there was also a significant difference in the scores for spasticity at baseline (M = 2.71, SD = 0.76) and after the SCS therapy (M = 2.14, SD = 0.69); Z(7) = 1.80, W = 10.00, *p* = 0.036. There was only one patient who did not respond to therapy according to MAS, namely, P06. The best result, which was a 3-point reduction in spasticity, was seen in the patient with spasticity in her arm only, namely, P07. All other patients had a 1-point decrease in spasticity in the arm. Four patients, namely, P03, P04, P05, and P07, showed a decrease in spasticity in the leg. Only one patient (P11) in the control group had a slight decrease in both arm and leg spasticity from 2 points to 1 point during the observation period. The rest of the patients had no changes in their Ashworth scores.

### 3.2. Seed-Based Functional Connectivity Analysis 

Functional connectivity maps were generated by computing the correlation coefficient between the average time series in the seven seed areas: brainstem, left and right precentral, postcentral and supplementary motor cortices, and the rest of the brain, and then normalized to Fisher-*Z* values for comparison. 

From the variance analysis of the difference between post- and pre-therapy connectivity measures, a significant effect of the group factor (p_FDR-corr_ < 0.05) was found in several seed brain regions. The list of these structures is given in Table 3. These were the brainstem, right pre- and postcentral cortex, and the supplementary motor area.

The post hoc pairwise comparison results show that following the course of SCS therapy, there was a significant increase in the functional connectivity with the seed in the brainstem in one cluster of 669 voxels (Figure 4A, Table 4), with the center of mass at (x = 18, y = −12, z = 74) that contained, for the most part, the right premotor cortex (BA6). In contrast, the control group showed a decrease in functional connectivity of the brainstem with the right temporal pole and the right orbital part of the inferior frontal gyrus (Figure 5A, Table 4).

In the main group, we also found an increase in functional connectivity with the seed in the right precentral gyrus and a single cluster of 376 voxels, with the center of mass at (x = 14, y = −16, z = 46) corresponding mostly to the anterior cingulate gyrus (Figure 4B, Table 4). No changes in the functional connectivity of this area were observed in the control group.

A seed in the right supplementary motor area showed a significant increase in functional connectivity after the SCS therapy, with the cluster of 360 voxels (center of mass at x = −38, y = −24, z = 46) located predominantly in the left postcentral gyrus (BA1) (Figure 4C, Table 4). In the control group, no significant changes in functional connectivity were obtained.

Reduced functional connectivity following the SCS course was found between the seed in the left postcentral gyrus and one cluster (size—722, center of mass (x = −30, y = −56, z = 50)), mainly consisting of the left parietal cortex (BA7) (Figure 4D, Table 4). A similar analysis performed for the control group showed no changes in functional connectivity for this seed. 

In the control group, there was a decrease in the functional connectivity of the right postcentral gyrus with the central regions of the cerebellum (Figure 5B, Table 4). No significant effects in the right postcentral gyrus as a seed were observed in the main group. 

The left precentral and the left supplementary motor area did not show any statistically significant effect either in the main or in the control group.

### 3.3. ROI-to-ROI Analysis

ROI-to-ROI analysis of resting state functional connectivity was conducted using seeds in the same regions above and the sensorimotor network, in addition to the seed-to-voxel analysis. 

The comparative analysis performed within the subnetwork (a cluster of connections) consisting of seven regions of interest within the motor areas did not identify a significant change in functional connectivity at the ROI-to-ROI level. 

The analysis performed for the connections within the sensorimotor network revealed an increased connection between the right and left lateral nodes following the SCS therapy (*p* = 0.005, FDR-corrected; cluster-defining threshold at *p* < 0.01, uncorrected). In the control group, no changes in functional connectivity were obtained in a similar comparison.

### 3.4. Whole-Brain Correlation Analysis

Since we observed a significant decrease in spasticity in the main group, we wondered which patterns of change in functional connectivity were associated with this clinical improvement. To find the functional brain correlates of the elimination of spasticity syndrome, we performed a search analysis of whole-brain functional connectivity. In order to answer the question of whether the correlation between connectivity change and spasticity recovery differed from zero, we constructed a post-SCS minus pre-SCS between-conditions contrast and used deltas of the post- minus pre-SCS MAS values as the between-subject contrast. We found no correlation of functional connectivity with reduced spasticity values in the hand. 

There was a positive correlation between the spasticity decrease in the leg (the higher the delta MAS score, the stronger the mitigation of spasticity) and the functional connectivity increase in two pairs of regions: (1) right superior sensorimotor cortex (0, −31, 67)–right postcentral gyrus (R^2^ = 0.95, β = −0.68, two-sided *p* = 0.0011; Figure 4), and (2) right superior sensorimotor cortex (0, −31, 67)–right superior parietal lobule (R^2^ = 0.86, β = −0.89, two-sided *p* = 0.008; Figure 6). 

In summary, the greater antispastic effect in the leg was associated with an increase in intrahemispheric connections of the middle node of the sensorimotor network and the cortical motor areas nearest to it following the course of SCS treatment.

## 4. Discussion

In the present pilot study, it was shown that epidural spinal cord stimulation for a short period resulted in functional changes within the central motor system compared with the controls. The first was higher functional connectivity of the brainstem to the premotor cortex in the right hemisphere, as well as changes in interhemispheric functional connectivity between cortical motor areas; the second was an increase in functional connectivity between the right and left lateral nodes of the sensorimotor network; the third was a positive correlation between the decrease in spasticity and an increase in functional connectivity within the right hemisphere sensorimotor cortex. All these changes in functional connectivity occurred with a statistically significant decrease in spasticity, as assessed by the MAS. Interestingly, in the control group, we did not observe any clinical changes in terms of decreased spasticity during this observation period. Moreover, patients in the control group demonstrated only a decrease in functional connectivity. These reductions were observed in the area of the brainstem with the right temporal pole and the right orbital part of the inferior frontal gyrus, as well as in the right postcentral gyrus with the central regions of the cerebellum.

Following the short course of SCS therapy, we first observed an increase in resting-state functional connectivity between the brain stem and the contralesional premotor cortex (BA6) (which is the major site of origin of the corticoreticular tract) of the right hemisphere. Given the pathogenetic mechanisms of a UMN lesion, this finding looks very intriguing. A recent study has shown that post-stroke spasticity as measured by diffusion-tensor imaging (DTI) is associated with decreased fractional anisotropy in the contralesional corticoreticular pathway [59]. The increase in functional connectivity between the brainstem and cortical motor area obtained in our study indicates a more synchronized functioning of these two areas. Thus, our findings may, first, provide evidence for a direct therapeutic effect of epidural cervical spinal cord stimulation in patients with acquired spasticity, and second, clarify the recovery mechanisms present in anti-spastic SCS. This mechanism clearly corresponds to recovery through the transcallosal pathway, one of three suggested recovery mechanisms for an injured corticoreticular tract (recovery through the original pathway, recovery through perilesional reorganization, and recovery through the transcallosal pathway) [60].

Changes in functional connectivity between cortical motor areas were bidirectional. Between the right precentral gyrus and right cingulate cluster; right supplementary motor cortex and the left postcentral gyrus (BA1) we observed an increase in functional connectivity after the short course of SCS therapy. In contrast, between the left postcentral gyrus and the left parietal cortex cluster (BA7), we observed a drop in functional connectivity at the second control point. More generally, there were three main changes: (1) an intrahemispheric increase in the functional connectivity between the motor areas and the anterior cingulate, (2) an increase in interhemispheric functional connectivity between the right motor areas and left postcentral cortex, and (3) an intrahemispheric decrease in connectivity between the postcentral gyrus and left parietal cortex. 

Neurons in the anterior cingulate gyrus seem likely to play some role in controlling the muscle stretch reflex. There is documented evidence of two cases of ischemic stroke entailing the anterior cingulate gyrus (pericallosal artery territory) that led to the development of spasticity [61]. The precentral cortex is the anatomical location of the primary motor cortex. 

The left precentral gyrus is involved in voluntary motor movement on the body’s right side. A total of 80% of the corticospinal neurons originate in the precentral cortex [62]. There is evidence in the literature that damage to the supplementary motor cortex and premotor cortex and the fibers from these corticoreticular and corticobulbar tracts, respectively, can lead to spasticity [62,63]. In light of this knowledge, it is possible to conclude that the enhanced right supplementary motor cortex–left postcentral gyrus relationship obtained in our work played an adaptive role in the regulation of right limb muscle tension by the supplementary motor cortex. 

It is difficult to say whether the decline in intrahemispheric connectivity of the primary and secondary motor cortices is an adaptive or maladaptive rearrangement. We lean toward the former and will try to base our assumptions on two postulates. First, the parietal area, together with the frontal areas, form a parieto-frontal network that implements goal-oriented movements, including reach-to-touch and reach-to-grasp movements, in humans [64] and monkeys [65]. This network has a large number of intrahemispheric connections, including those between the parietal and primary motor cortices [66,67,68]. And it is obvious that the implementation of purposeful movement is impossible without online somatosensory feedback. Second, patients with spasticity often have a co-contraction of the muscles around the joint, which may be a strategy to overcome inadequate prediction of the sensory consequences of movements [69,70,71]. The insufficient prediction of somatosensory feedback from the moving limb may result in compensatory or even maladaptive hyperconnectivity between somatosensory and planning areas, which weakens as spasticity decreases. 

Cortical remodeling following treatment of spasticity with more traditional treatments, such as botulinum toxin type A, also occurs in post-stroke patients and is accompanied by significant changes in functional connectivity in the contralesional sensorimotor and bilateral parietal cortex. These changes were detected over 4–11 weeks between scans, which was much longer than in our study [72]. Cortical rearrangements, mainly in areas related to motor function, detected using fMRI during a motor task with spasticity treatment have also been observed in other studies [73,74,75,76]. Thus, our results are in line with both the data on the pathogenesis of spasticity and the data on neuronal rearrangements with spasticity treatment. 

This pilot study also suggests increased functional connectivity within the sensorimotor network between the right and left lateral nodes. There is evidence in the literature that interhemispheric interactions within the sensorimotor network are impaired after a stroke with motor deficits [77,78]. Our findings may indicate the reintegration of the sensorimotor network nodes after a course of anti-spastic SCS therapy. 

The use of resting-state analysis also aimed to determine whether there was any connection between the clinical improvement and the functional connectivity rearrangement. For this purpose, we used a hypothesis-free whole-brain analysis of the correlation of Ashworth scale improvements in leg and hand with all functional connections between 165 ROIs covering all brain structures. After adjusting for multiple comparisons, only two pairs of functional connections were found to correlate with decreased MAS scores in the leg only. The pairs were the right medial sensorimotor cortex–right postcentral gyrus and right medial sensorimotor cortex–right superior parietal lobule. Both correlations were positive, that is, clinical improvement was greater with a greater increase in connectivity in these pairs, or the increase in connectivity between these areas promoted the reduction in spasticity. This increase in functional connectivity can only be attributed to adaptive neuroplasticity, as it corresponded to greater spasticity mitigation (i.e., a greater difference in MAS scores). These results may indicate the onset of movement of function to the healthy hemisphere. Accordingly, our result complements the available data on the mechanisms of adaptive neuroplasticity, according to which an increase in functional lateralization toward the lesioned hemisphere corresponds to a favorable course of recovery [79].

Considering the clinical relevance of the present study, it should be said that even after a short exposure to epidural antispastic spinal cord stimulation at the cervical level, we observed a reduction in spasticity by an average of 1 point on the Ashworth scale. It is particularly encouraging that two of our patients experienced complete rigidity in flexion and extension (4 points) before therapy that was significantly alleviated after therapy (3 points). Another five patients with spasticity scores of 2–3 also experienced clinical improvement. And only one patient did not respond to the therapy. The combination of such encouraging clinical findings with the neuroimaging data obtained may indicate a potentially high clinical significance of the procedure. This pilot study also provides some new evidence of a possible therapeutic effect and cortical reorganization after anti-spastic SCS treatment. 

There were several limitations to the currently described investigation. First of all, the study by design could not address the question of whether the observed changes were part of spontaneous recovery or a placebo effect, or whether these neural processing differences were a consequence of the SCS therapy. Instead, the reported effects were longitudinal changes within patients following a short course of SCS therapy. It must be said that the effect of spontaneous recovery was counteracted by the duration of illness in the group: half of the patients had an illness duration between 190 and 3794 days; only two post-stroke patients were in the subacute stage when the most pronounced neuroplastic changes are possible, including spontaneous changes; and the remaining two post-traumatic patients were in the chronic phase of TBI. Also, spontaneous reduction in spasticity appears to be rare [80]. The placebo effect could not be controlled at this stage of the project, but it seems implausible to us to attribute the observed changes, given their strength, to this cause. Finally, it should be noted that the present findings are consistent with the literature documenting the pathogenesis of spasticity and the course of adaptive plasticity. 

Due to the resource-intensive nature of the procedure, we were only able to recruit 8 patients with left-sided cerebral lesions and, consequently, right-sided spasticity in the arm and leg. Given the small sample size, the generalizability of our findings is limited and there may be a lack of power to detect additional behavioral/functional relationships. Nevertheless, we chose to control for multiple comparisons of connectivity and brain-symptom relation analyses. We believe the fact that there was a statistically significant effect in such a small sample size argues, albeit indirectly, for the presence of a significant neuromodulatory effect of SCS. A larger sample size may have revealed additional significant findings; however, despite the small sample size, significant clinical and functional brain changes were observed. Nonetheless, future studies with larger sample sizes and across various medical centers are warranted to better understand how SCS therapy impacts the brain and eliminates spasticity.

Usually, in studies of plastic changes in the brain after cerebral accidents, patients are selected homogeneously according to the duration of the disease, and there is evidence that functional re-organization may follow different pathways across disease phases, but in our study, we did not have this luxury. However, the heterogeneity of the group according to disease duration in this design played to our advantage to some extent, as heterogeneity should logically eliminate possible effects associated with the duration after the stroke. However, we do not deny the need for clearer criteria in further stages of this clinical trial.

Also, a weakness of our study was that no fMRI with a task was performed. Although motor paradigms could somewhat improve the accuracy of motor area identification, we focused on existing atlases of functional cortical areas. 

The short period of test stimulation in our study was determined, first, by the limitation of the duration of hospitalization in our center and, second, by the recommendations of the electrode manufacturer. For the same reasons, we could not manage two different time points before SCS; according to the established protocol, patients went to the operation to install the system almost immediately after the beginning of hospitalization. 

Also, because of the extremely limited hospitalization time, we did not perform follow-up RS-fMRI to assess the duration of the therapeutic effect and the rate of its washout. This issue is obviously extremely important and we will do our best to study it in the future.

Our future work will focus on (1) increasing the statistical power by increasing the sample size and homogeneity, (2) increasing the sensitivity in assessments of (increased) muscle tension, and (3) increasing the anatomical accuracy of the neuroimaging data and analysis.

## 5. Conclusions

The measurements of brain connectivity at baseline and after the SCS revealed neuroplasticity associated with mitigation of spasticity. Enhanced functional connectivity between the contralesional premotor cortex and brainstem, which are, respectively, the beginning and the end of the corticoreticular tract, has been shown to play a leading role, according to the dominant view, in the pathogenesis of spasticity. Modulation of intra- and interhemispheric cortico-cortical interactions between primary and secondary motor areas was also shown. Reduced spasticity was associated with increased contralesional interhemispheric functional connectivity after the SCS. Our findings suggest a possible therapeutic mechanism of action of the SCS in the treatment of acquired spasticity. It also describes functional connectivity rearrangements after a short course of SCS, which may contribute to an understanding of the mechanisms of adaptive neuroplasticity of the motor system.

## Figures and Tables

**Figure 1 biomedicines-11-02266-f001:**
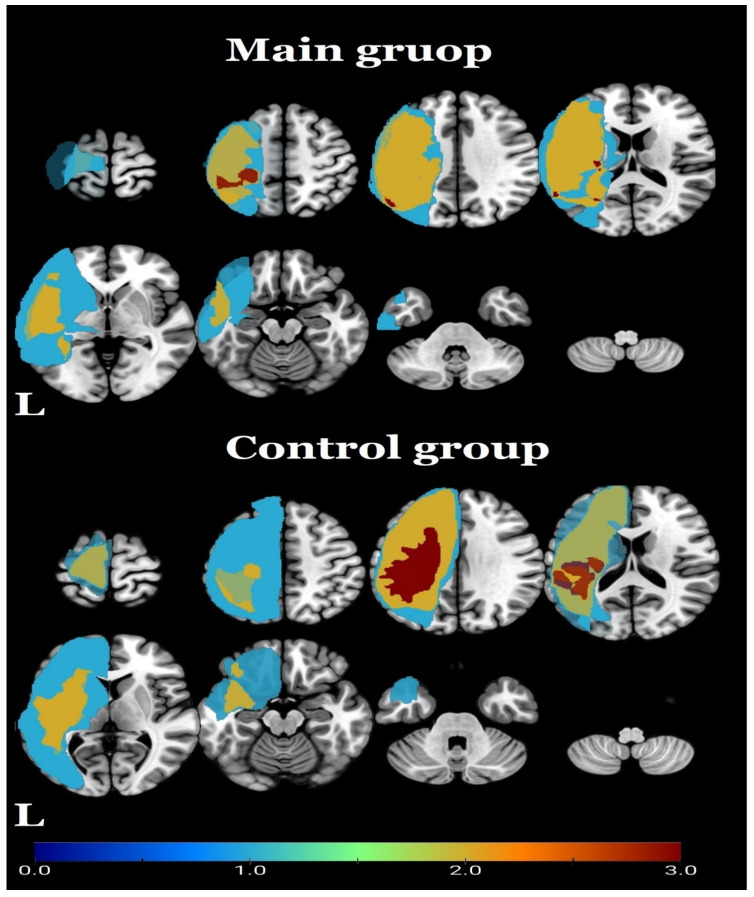
Heatmap of lesion localization of the study population on an MNI standard brain template. The legend shows the number of lesions that are represented by the colors in the figure.

**Figure 2 biomedicines-11-02266-f002:**
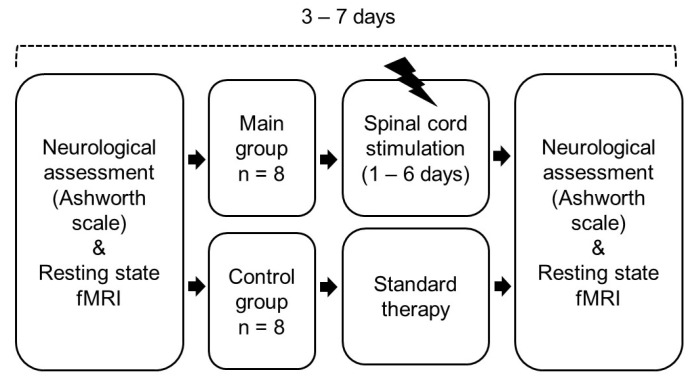
Flowchart of patient selection.

**Figure 3 biomedicines-11-02266-f003:**
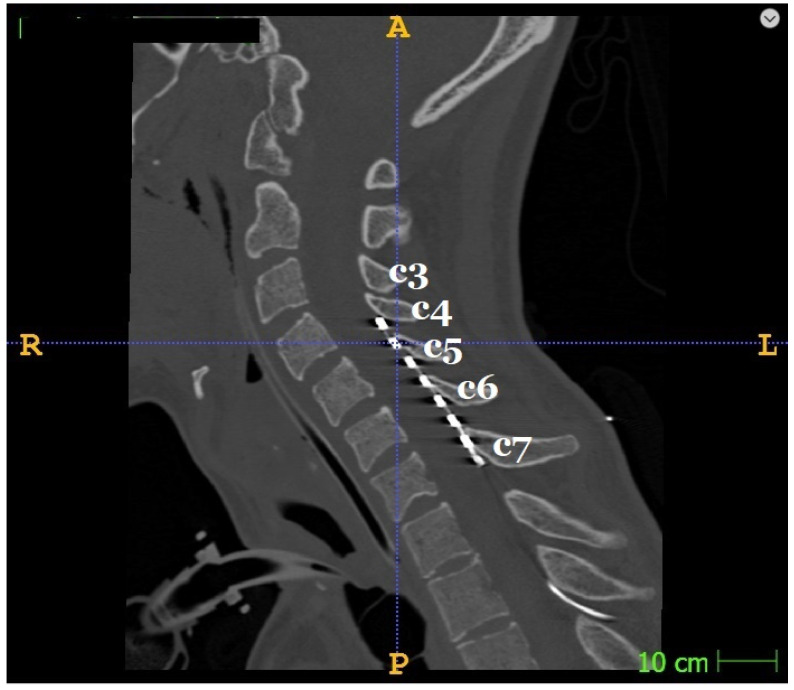
CT control of an electrode position (example, patient P05). C3–C7 marks the spinous processes of the cervical vertebrae 3–7.

**Figure 4 biomedicines-11-02266-f004:**
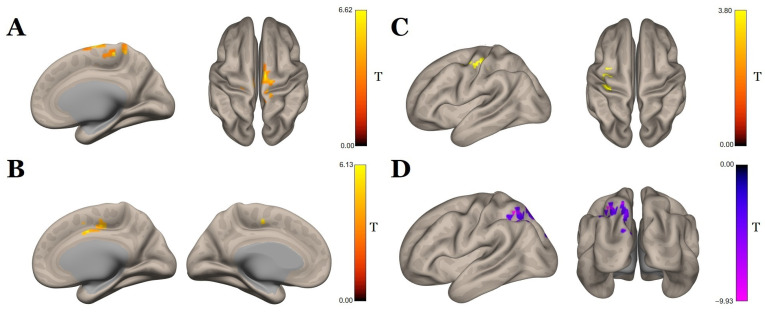
Seed-to-voxel analysis, t-statistics for changes (post-SCS vs. pre-SCS) in functional connectivity in the main group with the seed in the (**A**) brainstem, (**B**) right precentral gyrus, (**C**) right supplementary motor area, and (**D**) left postcentral gyrus (*p* < 0.05, FDR-corrected); color bars represent positive t-values in orange and negative t-values in blue.

**Figure 5 biomedicines-11-02266-f005:**
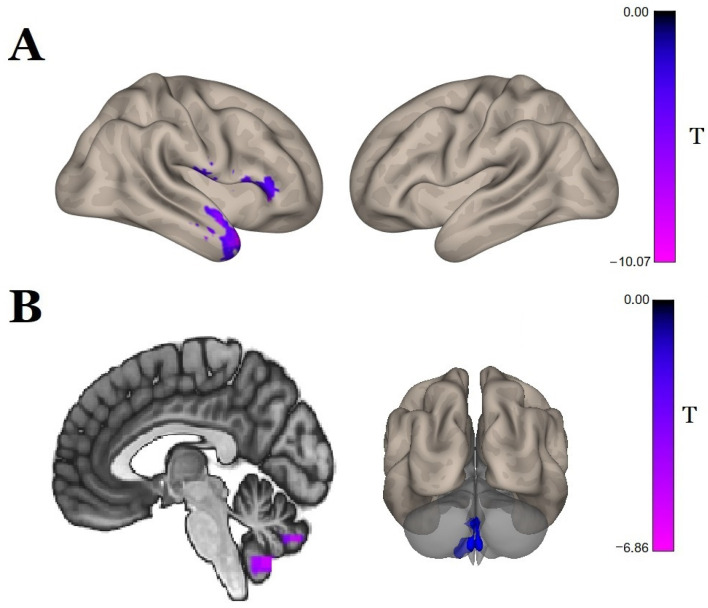
Seed-to-voxel analysis, t-statistics for changes (2nd control point vs. 1st control point) in functional connectivity in the control group with the seed in the (**A**) brainstem and (**B**) right postcentral gyrus (*p* < 0.05, FDR-corrected); color bars represent negative t-values in blue.

**Figure 6 biomedicines-11-02266-f006:**
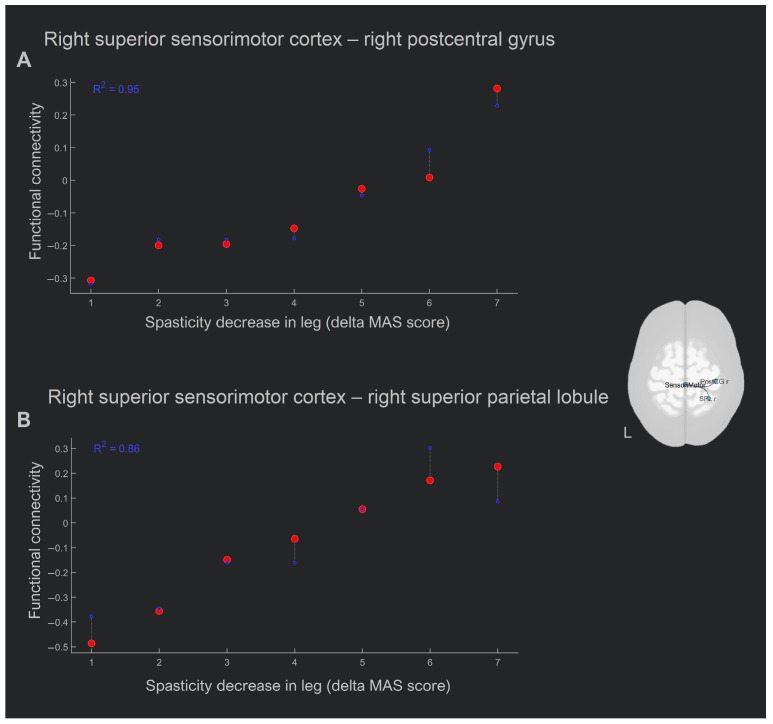
Whole-brain analysis results, where a positive correlation of spasticity decreases with the difference (post-SCS minus pre-SCS) in functional connectivity in the (**A**) right superior sensorimotor–right postcentral cortex and (**B**) right superior sensorimotor–right superior parietal cortex pairs (*p* < 0.05, FDR-corrected). Abbreviation key: l—left, r—right.

**Table 1 biomedicines-11-02266-t001:** Demographical, clinical, and data of intervention outline.

Patient	Age	Sex	Handedness	Etiology	Lesion Side	Brain Tissue Damage Localisation	Lesion Volume (cm^3^)	Time since Stroke (days)	AS Hand	AS Leg	SCS Duration Days (hours per day)	SCS Mode	Time between fMRI Sessions (days)
	*Main group*												
P01	35	M	R	Hemorrhage	L	CS	284.18	703	4	3	4 (12)	Tonic	7
P02	50	F	R	IS	L	CS	157.99	190	2	2	3 (12)	Tonic	7
P03	66	M	R	Hemorrhage	L	S	2.44	81	2	2	5 (12)	Tonic	5
P04	51	M	R	IS	L	CS	29.74	22	3	3	5 (12)	Burst	5
P05	25	F	R	Anoxic	-	-	-	1426	3	3	5 (12)	Burst	6
P06	50	M	R	TBI	L	C	15.92	65	2	2	1 (12)	Tonic	4
P07	19	M	R	TBI	L	CS	152.46	65	3	-	4 (12)	Burst	4
P08	32	F	R	TBI	L	S	0.46	3794	4	4	6 (12)	Tonic, burst	6
	*Control group*												
P09	57	M	R	IS	L	CS	176.58	269	-	-	-	-	5
P10	24	F	R	TBI	L	C	2.328	1094	4	3	-	-	5
P11	75	M	R	IS	L	CS	217.26	65	2	2	-	-	7
P12	58	M	R	IS	L	CS	48.65	524	-	-	-	-	3
P13	54	M	R	IS	L	S	78.86	2230	2	2	-	-	6
P14	66	M	R	IS	_	_	_	20	1	1	-	-	4
P15	55	M	R	IS	_	_	_	251	1	1	-	-	3
P16	63	F	R	IS	L	S	5.9	361	1	1	-	-	4

AS—Ashworth scale; C—cortical; CS—cortico-subcortical; F—female; IS—ischemic stroke; L—left; M—male; P—patient; R—right; S—subcortical; SCS—spinal cord stimulation; TBI—traumatic brain injury.

**Table 2 biomedicines-11-02266-t002:** Median values of the subject groups, interquartile range, the value of the Mann–Whitney U-test, and Wilcoxon signed-rank test (for spasticity rates) comparing the groups of subjects.

	Main Group (*n* = 8)	Control Group (*n* = 8)	*p*-Value
Age, years	42.5 (24)	56.5 (11)	0.046
Female, %	27.5	25	-
Disease duration, days	135.5 (1180.25)	315 (840)	0.795
Lesion volume, cm^3^	22.8 (155.63)	27.8 (151.57)	0.873
AS hand	3 (2)	1 (4)	>0.05
AS leg	2.5 (4)	1 (3)	>0.05
Time between 1st and 2nd fMRI	5.5 (2.5)	4.5 (2.5)	0.226

**Table 3 biomedicines-11-02266-t003:** Resting-state functional connectivity, RM-ANOVA 2 × 2.

Seed	Region	MNI (x, y, z)	Cluster Size	F_(2,14)_	p-FDR
Brainstem	Temporal pole r	52, 8, −32	725	32.02	0.000
	Angular g. r	54, −64, 38	250	28.19	0.000
	Orbital part of inferior frontal g. r	46, 30, 0	248	18.38	0.000
Precentral g. r	Anterior prefrontal cortex r	10, 46, 12	313	22.24	0.000
Precentral g. l	No significant effect				
Postcentral g. r	Cerebellum	−2, −62, −50	427	25.86	0.000
Postcentral g. l	No significant effect				
Supplementary motor cortex r	No significant effect				
Supplementary motor cortex l	Cerebellum	−12, −50, −56	84	46.96	0.000

**Table 4 biomedicines-11-02266-t004:** Seed-to-voxel analysis, brain regions showing alterations to functional connectivity with selected seeds in patients of the main group (contrast after SCS therapy > before SCS therapy) and control group (contrast 2nd control point > 1st control point). Cluster location, MNI co-ordinates (x, y, z), and T maxima (cluster-level FDR-corrected) are shown.

Seed	Region	BA	MNI (x, y, z)	Cluster Size	T	p-FDR
*Main group*						
Brainstem	Premotor cortex r Supplementary motor cortex r	6	18, −12, 74	669	9.19	0.007
Precentral g. r	Ventral anterior cingulate cortex r	24	14, −16, 46	376	6.70	0.049
Postcentral g. l	Visuo-motor area l	7	−30, −56, 50	722	−12.76	0.000
Supplementary motor cortex r	Primary somatosensory cortex l	1	−38, −24, 46	360	7.81	0.041
Precentral g. l	No significant effect					
Postcentral g. r						
Supplementary motor cortex l						
*Control group*						
Brainstem	Temporal pole r	38	46, 14, −28	1275	−8.75	0.000
	Orbital part of inferior frontal g. r	47	46, 28, −4	481	−5.23	0.001
Postcentral g. r	Cerebellum	-	0, −74, −40	662	−6.91	0.000
Precentral g. r	No significant effect					
Precentral g. l						
Postcentral g. l						
Supplementary motor cortex r						
Supplementary motor cortex l						

## Data Availability

The raw data supporting the conclusions of this article will be made available by the authors without undue reservation.

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
