# Peer review of "Immediate Effects of Anti-Spastic Epidural Cervical Spinal Cord Stimulation on Functional Connectivity of the Central Motor System in Patients with Stroke- and Traumatic Brain Injury-Induced Spasticity: A Pilot Resting-State Functional Magnetic Resonance Imaging Study"

_biomedicines, 2023, doi:10.3390/biomedicines11082266_

Round 1
Reviewer 1 Report
The effect of the anti-spastic C3-C5 spinal cord stimulation (SCS) on the activity of the supraspinal neural systems involved in the pathogenesis of spasticity was verified. Moreover, the study aimed to find out whether the effect of spasticity reduction alters the patterns of functional connectivity of the brain.
Eight patients with upper motor neuron (UMN) syndrome as a result of left hemispheric brain damage were assessed with fMRI immediately before and immediately after short-term cervical epidural SCS therapy. Eight patients with UMN but with conventional therapy were examined as a control group.
The authors concluded the positive effects of SCS therapy by means of the higher functional connectivity in supraspinal areas (subcortical and cortical), increased functional connectivity in sensorimotor cortex areas, and a positive correlation between decreased spasticity and increased functional connectivity within the right hemisphere sensorimotor cortex.
The paper is interesting and the results are a convincing novelty even though the number of performed tests (number of the studied subjects) is low.
In this article, there are several flaws, or rather understatements or editorial lacks, that should be clarified:
1. In Abstract it should be explained in the detail “the upper motor neuron (UMN) syndrome”. What in fact the patients were ill with? The etiology of the injury is important in understanding brain connectivity changes following applied therapy.
2. “Eight patients with UMN but with conventional therapy” similarly as above sound mysterious and more data should be provided in the Abstract about them.
3. In Introduction: line 40 - …”Spasticity - abnormal muscle tightness”… should be here and throughout the text replaced with “increased muscle tension”, otherwise it can be missed with hypotonia.
4. lines 53-58 – the pathophysiology mechanism of the spasticity needs better explaining, what about the upper motoneurone dysfunction? Maybe rewriting data in lines 53-80 would give better effect.
5. The paper by Nagel et al. on SCI direct effects in treatment of spasticity is not enough and needs upgrading.
6. Table 1 and 4 are poorly presented.
7. Discussion should be started from “brief” important findings presentation.
8. Please provide more clinical relevancies coming from this study in the Discussion section. (…These changes were recorded for the first time, and given what is known about the pathogenesis of spasticity in upper motor neuron syndrome, and adaptive neuroplasticity, they may reflect a therapeutic effect of the SCS-therapy.”… - is not enough)
9. Citation in the text is not in accordance to the MDPI style, refs. list as well – very confusing.
10. Lack of ethical declaration at the end of the article and the authors' contribution.
Minor corrections
Author Response
Response to Reviewer 1 Comments
Dear reviewer, we appreciate your insightful comments and thank you for very thorough reading our manuscript. We tried to take all of them into account, what, we believed, resulted in the considerable improvement of the article. Following the reviewer suggestions and comments we corrected the text.
Point 1: In Abstract it should be explained in the detail “the upper motor neuron (UMN) syndrome”. What in fact the patients were ill with? The etiology of the injury is important in understanding brain connectivity changes following applied therapy.
Response 1: We clarified the etiology of brain injury in the main and control groups in abstract (lines 18-24). We have also removed the mention of UMN syndrome from the abstract. Since UMNS includes too wide spectrum of pathologies and CNS lesions at different levels, we tried to avoid this term by specifying the etiology of the disease, also in the title.
Point 2: “Eight patients with UMN but with conventional therapy” similarly as above sound mysterious and more data should be provided in the Abstract about them.
Response 2: Information on the control group has been clarified.
Point 3: In Introduction: line 40 - …”Spasticity - abnormal muscle tightness”… should be here and throughout the text replaced with “increased muscle tension”, otherwise it can be missed with hypotonia.
Response 3: Thank you for the valuable correction, corrections made.
Point 4: lines 53-58 – the pathophysiology mechanism of the spasticity needs better explaining, what about the upper motoneurone dysfunction? Maybe rewriting data in lines 53-80 would give better effect.
Response 4: Dear reviewer, thank you for the very useful correction! We have rewritten the data in lines 53-80, drawing on a number of recent articles.
Point 5: The paper by Nagel et al. on SCI direct effects in treatment of spasticity is not enough and needs upgrading.
Response 5: We have moved the reference to this author to a more appropriate context, where other studies on the anti-spastic effects of epidural SCS therapy are also cited. We checked the literature for new articles on the effect of epidural spinal cord stimulation on spasticity and found no new articles. Because we may have missed something, we would be very grateful if the reviewers could advise us of these articles.
Point 6: Table 1 and 4 are poorly presented.
Response 6: We have made Tables 1 and 4 more readable, if that is not enough, we can divide each of them into 2 (for the main group and the control group). We have also added information about the brain tissue damage localisation into Table 1.
Point 7: Discussion should be started from “brief” important findings presentation.
Response 7: We've removed all the unnecessary stuff from the beginning of the discussion and reworded the beginning a bit. Now the first paragraph of the Discussion has been changed to “In the present pilot study, it was shown that the epidural spinal cord stimulation for a short period of time results in functional changes within the central motor system compared to controls. The first was higher functional connectivity of the brainstem to the premotor cortex in the right hemisphere, as well as changes in interhemispheric func-tional connectivity between cortical motor areas; the second was an increase in func-tional connectivity between the right and left lateral nodes of the sensorimotor network; the third was a negative correlation between the decrease in spasticity and an increase in functional connectivity within the right hemisphere sensorimotor cortex. All these changes in functional connectivity occurred with a statistically significant decrease in spasticity as assessed by the MAS. Interestingly, in the control group, we did not observe any clinical changes in terms of decreased spasticity at this observation period. Moreover, patients in the control group demonstrated only a decrease in functional connec-tivity. These reductions were observed in the area of the brainstem with the right tem-poral pole and the right orbital part of the inferior frontal gyrus, as well as in the right postcentral gyrus with the central regions of the cerebellum.”
Point 8: Please provide more clinical relevancies coming from this study in the Discussion section.
(…These changes were recorded for the first time, and given what is known about the pathogenesis of spasticity in upper motor neuron syndrome, and adaptive neuroplasticity, they may reflect a therapeutic effect of the SCS-therapy.”… - is not enough).
Response 8: Thank you for the helpful observation, in accordance with the correction we bring the discussion:" Considering the clinical relevance of the present study, it should be said that even after a short exposure to epidural antispastic spinal cord stimulation at the cervical level, we observed a reduction in spasticity by an average of 1 point on the Ashworth scale. It is particularly encouraging that two of our patients experienced complete rigidity in flexion and extension (4 points) before therapy, which was significantly alleviated after therapy (3 points). Another 5 patients with spasticity scores of 2-3 also experienced clinical improvement. And only 1 patient did not respond to therapy. The combination of such encouraging clinical findings with the neuroimaging data obtained may indicate a potentially high clinical significance of the procedure. This pilot study also provides some new evidence of a possible therapeutic effect and cortical reorganization after an-ti-spastic SCS treatment.”
Point 9: Citation in the text is not in accordance to the MDPI style, refs. list as well – very confusing.
Response 9: We have changed the style of links to match the journal's requirements.
Point 10: Lack of ethical declaration at the end of the article and the authors' contribution.
Response 10: We entered relevant information from the minutes of the ethics committee meeting.

Reviewer 2 Report
The authors investigate and detect intrinsic brain connectivity rearrangements that occurred during spasticity mitigation, following short epidural SCS-therapy. The study is so interesting, however, I have some concerns to discuss.
What is the effect of SCS for peripheral nerve?
What is the side effect of SCS?
What is the mechanism of SCS?
Author Response
Response to Reviewer 2 Comments
Dear reviewer, thank you for your comments!
The authors investigate and detect intrinsic brain connectivity rearrangements that occurred during spasticity mitigation, following short epidural SCS-therapy. The study is so interesting, however, I have some concerns to discuss.
Point 1: What is the effect of SCS for peripheral nerve?
Response 1: Unfortunately, we did not find any information on the effect of cervical epidural stimulation on peripheral nerves. We may have misunderstood your question. In the answer to your third question, we give possible mechanisms of the effect of this type of therapy on the spinal level. Possible mechanisms of influence of the procedure on the supraspinal level are described in lines 313-329.
Point 2: What is the side effect of SCS?
Response 2: Thank you, we have included in the Introduction the data from a systematic analysis of literature of reported complications of epidural spinal cord stimulation (lines 305-308). We also indicated in the relevant section on methodology that none of our patients had complications during the entire study period.
Point 3: What is the mechanism of SCS?
Response 3: We've added information (lines 310-313).

Round 2
Reviewer 2 Report
The authors replied well ,so the manuscript sis suitable for publication.